# The novel microRNAs hsa-miR-nov7 and hsa-miR-nov3 are over-expressed in locally advanced breast cancer

Deepak Poduval[1], Zuzana Sichmanova[1], Anne Hege Straume[1¤], Per Eystein Lønning[1,2], Stian Knappskog[1,2]*

**1** Section of Oncology, Department of Clinical Science, University of Bergen, Bergen, Norway, **2** Department of Oncology, Haukeland University Hospital, Bergen, Norway

¤ Current address: Norwegian Institute of Marine Research, Bergen, Norway.
* stian.knappskog@uib.no

## Abstract

miRNAs are an important class of small non-coding RNAs, which play a versatile role in gene regulation at the post-transcriptional level. Expression of miRNAs is often deregulated in human cancers. We analyzed small RNA massive parallel sequencing data from 50 locally advanced breast cancers aiming to identify novel breast cancer related miRNAs. We successfully predicted 10 novel miRNAs, out of which 2 (hsa-miR-nov3 and hsa-miR-nov7) were recurrent. Applying high sensitivity qPCR, we detected these two microRNAs in 206 and 214 out of 223 patients in the study from which the initial cohort of 50 samples were drawn. We found hsa-miR-nov3 and hsa-miR-nov7 both to be overexpressed in tumor versus normal breast tissue in a separate set of 13 patients ($p = 0.009$ and $p = 0.016$, respectively) from whom both tumor tissue and normal tissue were available. We observed *hsa-miR-nov3* to be expressed at higher levels in ER-positive compared to ER-negative tumors ($p = 0.037$). Further stratifications revealed particularly low levels in the her2-like and basal-like cancers compared to other subtypes ($p = 0.009$ and $0.040$, respectively). We predicted target genes for the 2 microRNAs and identified inversely correlated genes in mRNA expression array data available from 203 out of the 223 patients. Applying the KEGG and GO annotations to target genes revealed pathways essential to cell development, communication and homeostasis. Although a weak association between high expression levels of *hsa-miR-nov7* and poor survival was observed, this did not reach statistical significance. *hsa-miR-nov3* expression levels had no impact on patient survival.

## Introduction

miRNAs are an important class of small non-coding RNAs, playing a versatile role in the gene regulation at the post–transcriptional level [1–5]. These molecules have proven to be involved in vital cellular functions, such as development, differentiation and metabolism [6–8]. In recent years there has been increased focus on the role of miRNAs in cancer [9], and the

**Data Availability Statement:** All relevant data are within the manuscript and its Supporting Information files and the raw data are available

through the Gene Expression Omnibus (accession: GSE145151).

**Funding:** This work was performed in the Mohn Cancer Research Laboratory. The project was funded by grants from the Trond Mohn Research Foundation, The Norwegian Cancer Society, The Norwegian Research Council and the Norwegian Health Region West.

**Competing interests:** The authors have declared that no competing interests exist

implementation of next generation sequencing (NGS) has led to the identification of multiple novel miRNAs as well as linked individual miRNA expression and combined signatures to tumor characteristics [10]. Currently there are 2656 distinct human miRNAs identified in the miRbase v22 [11], including more than 700 found to be deregulated in cancers [12].

Breast cancer is the most common malignancy in women. While outcome has improved significantly over the last three decades, resistance to therapy still presents a major challenge causing breast cancer related deaths [13]. As for chemoresistance in general, the underlying biological mechanisms remain poorly understood [14].

Merging evidence has indicated miRNA deregulation to play a role in breast cancer biology and outcome. Dysregulation of miRNAs may affect signal transduction pathways by targeting oncogenes and tumor suppressor genes [15], important to cancer development, progression, metastasis and potentially therapy response [16, 17]. Thus, while miR-10b, miR-125b, and miR-145 are generally downregulated, other miRNAs, like miR-21 and miR-155, are generally upregulated in breast cancer as compared to normal breast tissue [18]. Further, several miR-NAs have revealed strong associations to clinical parameters [19, 20]: For example, differential expression of miR-210, miR-21, miR-106b$^*$, miR-197, miR-let-7i, and miR-210, have been identified as a signature with prognostic value and also linked to invasiveness [21]. Moreover, miR-21 has been found linked to breast cancer metastasis and poor survival [22], while mir-29a overexpression has been shown to reduce the growth rate of breast cancer cells [23]. Given that many of the observed miRNA alterations are strongly cancer specific, this has inspired investigations into the potential use of miRNA as diagnostic biomarkers. Since miRNA are relatively stable molecules, they may be particularly attractive biomarkers to screen for in liquid biopsies (for original references, see [24])

miRNAs are also known to be differentially regulated across different subclasses of breast cancer. E.g. while members of the mir-181 family are up regulated in breast cancer in general, miR-181c in particular is activated by the expression of HER2 gene [25]. Also, miR-140 has been found suppressed by estrogen stimulation in ERα-positive breast cancer cells, most likely due to ER response elements in the flanking element of the miR-140 promoter [26].

In the present study, we analyzed global miRNA expression in 50 locally advanced breast cancers using NGS, aiming to identify novel, potentially breast cancer specific miRNAs. We identified and validated two novel miRNAs (one not previously described and one not previously reported in breast cancer), and subsequently evaluated their expression in an extended patient series (n = 223), by high sensitivity qPCR. Both were found over-expressed in breast cancer as compared to normal breast tissue. Considering different breast cancer subtypes, *hsa-miR-nov3* was expressed at particular high levels in ER-positive tumors contrasting lower levels in basal-like and Her2-like tumors. No similar patterns were observed for *hsa-miR-nov7*.

## Materials and methods

### Patients

In the present work we have analyzed biopsy material from two breast cancer studies.

1) In the first study, incisional biopsies were collected before chemotherapy from 223 patients with locally advanced breast cancer in a prospective study designed to identify the response to epirubicin (n = 109) and paclitaxel (n = 114) monotherapy. Primary response to therapy as well as long-term follow up (>10 years or death) was recorded for all patients. This cohort has been described in detail previously [27].

2) In the second study, tumor breast tissue and normal breast tissue from tumor bearing and non-tumor bearing quadrants were collected from 46 anonymous breast cancer patients

undergoing mastectomy, with the purpose of determining tissue estrogens. This cohort is described in detail in [28].

Using NGS, we analyzed miRNA expression in 50 patients from study 1). Next, candidate miRNAs were quantified using qPCR in all 223 patients from study 1), as well as 13 randomly selected patients from study 2), where RNA was available from tumor tissue and matching normal breast tissue (7 ER-positive and 6 ER-negative tumors). In addition, mRNA expression array data was available for 203 out of the 223 patients in study 1).

All patients provided written informed consent, and the studies conducted in accordance to national laws, regulation and ethical permissions (Norwegian health region West; REK Vest).

## Tissue sampling and RNA extraction

Tissue samples were snap-frozen in liquid nitrogen in the operating theatre and stored in liquid nitrogen until further processing. Total RNA was extracted from the biopsies using miRvanaTM kit (ThermoFisher), according to the manufacturer's instructions. RNA integrity and concentration were determined using Bioanalyzer 2000 and Nanodrop ND2000 spectrophotometer, respectively.

## miRNA-sequencing

Sample preparation and single-end sequencing were performed at the core facility of the Norwegian Genomics Consortium in Oslo, on Illumina HiSeq 2500, 1x50bp. De-multiplexing was performed using the Illumina CASAVA software. FastQC was run on all samples with the main purpose to assess sequence quality. The raw data are available through the Gene Expression Omnibus (accession: GSE145151).

## Novel miRNA prediction

The raw sequencing files (fastq) were processed using the novel miRNA prediction algorithm mirdeep v2.0.0.5 [10]. Potential novel miRNAs were identified using the human reference genome (hg19) and already identified miRNAs from humans and other hominids from miRbase 20 [29]. In the mirdeep2 algorithm, filtering parameters randfold P-value less than 0.05 and scores greater than or equal to 10 were applied. Precursor structures obtained after filtering were manually identified based on the presence of 1–2 mismatches in the stem region, a loop sequence of 4–8 nt, and the presence of mature sequence in the stem region (See S1 File.) [30].

## Validation of predicted novel miRNAs

Validation of the predicted novel miRNAs was performed by qPCR-based amplification of the miRNAs, with subsequent cloning and capillary sequencing of the products, to pinpoint the exact size and sequence of the miRNAs (see sections below for details).

## cDNA synthesis and qPCR

cDNA from miRNAs was prepared using Exiqon's Universal cDNA synthesis kit II, with 20 ng of total RNA as input. qPCR was performed using Exiqon's miRCURY LNA™ Universal RT microRNA PCR system, with custom Pick-&-Mix ready to use PCR plates with an inter-plate calibrator, on a LightCycler 480 instrument (Roche). Relative expression levels for each sample were calculated by dividing the expression of the gene of interest on the average expression of two reference miRNAs: miR-16-5p and miR-30b-5p.

## miRNA cloning and capillary sequencing

End products from custom miRNA specific qPCR were cloned into pCR 2.1 TOPO-TA vector (Life Technologies) by TOPO-TA cloning according to the manufacturer's instructions. The generated plasmids were amplified by transformation and cultivation of E. coli TOP10 cells (Life Technologies). The plasmids were then isolated using the Qiagen miniprep kit according to the manufacturer's instructions.

Sequencing was performed using the BigDye v.1.3 system (Applied Biosystems) and the primers following thermocycling conditions as previously described [31]. Capillary electrophoresis and data collection were performed on an automated capillary sequencer (ABI3700).

## Target prediction and pathway analysis

Target prediction was performed using the offline algorithm miRanda [32, 33] and the online algorithms miRDB [34] and TargetScanHuman Custom (Release 5.2) [35].

miRanda predicts gene targets based on position specific sequence complementarity between miRNA and mRNA using weighted dynamic programming, an extension of the Smith-Waterman algorithm [36]. Also, the miRanda algorithm uses the free energy estimation between duplex of miRNA: mRNA (Vienna algorithm [37]) as an additional filter.

The miRDB is an online database of animal miRNA targets, which uses SVM (Support Vector Machine) machine-learning algorithm trained with miRNA-target binding data from already known and validated miRNA-mRNA interactions [34, 38].

TargetScanHuman Custom predicts biological miRNA targets by searching for match for the seed region of the given miRNA that is present in the conserved 8-mer and 7-mer sites [35]. It also identifies sites with conserved 3' pairing from the mismatches in the seed region [39, 40].

An in-house pan-cancer panel of 283 tumor suppressor genes was used to filter target genes of interest. The panel was generated based on the tumor suppressors within the CGPv2/3-panels [41], Roche's Comprehensive Cancer Design as well as a manual literature search (S1 Table).

Further, we used GATHER, a functional gene enrichment tool, which integrates various available biological databases to find functional molecular patterns, in order to find biological context from the target gene list [42]. With the help of GATHER, we did KEGG pathway [43], and GO (gene ontology) enrichment analyses for the common genes predicted by all three prediction algorithms. Further, validations were performed using DAVID [44] and topGO [45].

## mRNA expression

In the interest of validating miRNA targets, we analyzed inverse correlations between miRNA expression and mRNA levels. mRNA expression levels were extracted from microarray analyses performed on a Human HT-12-v4 BeadChip (Illumina) after labeling (Ambion; Aros Applied Biotechnology). Illumina BeadArray Reader (Illumina) and the Bead Scan Software (Illumina) were used to scan BeadChips. Expression signals from the beads were normalized and further processed as previously described [46]. We re-annotated the data set using illuminaHumanv4.db from AnnotationDbi package, built under Bioconductor 3.3 in R [47], to select only probes with "Perfect" annotation[48]. The probes represented 21043 identified and unique genes (13340 represented by single probe and 7703 represented by multiple probes). In the cases of multiple probes targeting the same gene, we calculated fold difference for these probes. This was done to avoid losing potentially relevant biological information if expression of one probe was significantly higher that expression of another. However, for no genes did we find a fold difference >2 fold. Therefore, the mean expression for each such gene, was

calculated based on the values form each probe, weighted according to the number of beads per probe.

## Statistics

Expression levels of miRNAs in tumor versus normal tissue were compared by Wilcoxon rank tests for paired samples. Inverse correlations between miRNA expression and mRNA expression were assessed by Spearman tests. The potential impact of the novel miRNAs on long-term outcome (relapse-free survival and disease-specific survival) in breast cancer patients was calculated by Log-rank tests and illustrated by Kaplan-Meier curves, using the SPSS software v.19. All p-values are reported as two-sided.

## Results

### Novel miRNA prediction

In order to identify novel miRNAs, 50 patients with locally advanced breast cancer (from study 1, see materials and methods) were subject to global miRNA-sequencing using massive parallel sequencing. On average, the dataset resulted in 3 million reads per sample. Using the miRNA identifier module in miRDeep2, we detected 10 novel miRNAs (Table 1). Eight out of these 10 miRNAs were detected in a single sample only, while two were expressed in two or more patients and therefore regarded as the most reliable predictions. These two miRNAs, here temporarily named *hsa-miR-nov3* and *hsa-miR-nov7*, were found in tumor samples from 2 and from 6 patients, respectively. For both of these novel miRNAs, we identified precursor structures with not more than one or two mismatches in the stem region, as well as the presence of mature miRNA sequences (Fig 1; S1 Fig). Therefore, we selected these two miRNAs for further analyses. Cross-checking the miRCarta database [49], no hits were found for either of the two, but notably, while this work was conducted, *hsa-miR-nov7* was identified by another team in lymphomas, and reported as miR-10393-3p [50].

### *In-vitro* validation of novel micro RNAs

Next, we aimed to validate our *in-silico* predictions and confirm that the sequences from which we identified *hsa-miR-nov3* and *hsa-miR-nov7* represented bona-fide novel miRNAs expressed in the patients. Utilizing total RNA from the patients found to express the two predicted novel miRNAs, we performed global poly-adenylation and cDNA synthesis followed by miRNA-specific qPCR amplification. For both miRNAs we observed positive qPCR reactions. Further, end products of the qPCRs were then ligated into carrier-plasmids and sequenced.

**Table 1. Novel miRNA sequences as predicted by mirdeep v2.0.0.5 from massive parallel sequencing of total miRNA in 50 locally advanced breast cancers.**

| miRNA | Co-ordinate | Mature sequence | Strand | Number of samples |
|---|---|---|---|---|
| *hsa-miR-nov2* | chr2:36662749..36662809 | AAAAACTGCGATTACTTTTGCA | - | 1 |
| *hsa-miR-nov3* | chr3:186505088..186505149 | AAAGCAGGATTCAGACTACAATAT | + | 2 |
| *hsa-miR-nov3_2* | chr3:132393169..132393224 | CAAAAACTGCAATTACTTTTGC | + | 1 |
| *hsa-miR-nov4* | chr4:155140075..155140134 | AAAAGTAATCGCTGTTTTTG | + | 1 |
| *hsa-miR-nov7* | chr7:138728845..138728903 | AATTACAGATTGTCTCAGAGA | - | 6 |
| *hsa-miR-nov8* | chr8:116546693..116546762 | TTAGAGCTTCAACCTCCAGTGTGA | - | 1 |
| *hsa-miR-nov10* | chr10:31840034..31840078 | CGCGGGTGCTTACTGACCCT | + | 1 |
| *hsa-miR-nov10_2* | chr10:72163928..72163994 | GCGGCGGCGGCGGCGGCG | + | 1 |
| *hsa-miR-nov17* | chr17:36760852..36760906 | CCCAGCCCCACGCGTCCCCATG | - | 1 |
| *hsa-miR-nov20* | chr20:26189318..26189366 | TGGCCGAGCGCGGCTCGTCGCC | - | 1 |

We confirmed the resulting plasmids to contain the predicted miRNA sequences. Further, in both cases, the sequences were flanked by a poly A tail, confirming that the original molecules used as input in the poly-adenylation were present as short 22nt RNAs (Fig 2). Thus, we confirmed the presence of miRNAs with the exact sequence as predicted from the NGS-based data.

## Overexpression of hsa-miR-nov7 and hsa-miR-nov3 in breast cancer

Given that the sensitivity for the novel miRNAs was better in the qPCR than in the miRNA massive parallel sequencing (MPS) analysis, we aimed to assess whether the miRNAs were expressed in a limited number of breast cancer patients only (as indicated by their detection in 2 and 6 out of 50 patients in the MPS analysis), or if they were detectable in a higher fraction of patients, when applying a more sensitive detection method. We therefore performed qPCR to quantify the expression levels of *hsa-miR-nov7 and hsa-miR-nov3* in tumor tissue across the entire cohort of patients from study 1 (n = 223). With this method, we detected *hsa-miR-nov7* and *hsa-miR-nov3* in 206 and 214 samples out of total 223 samples respectively, albeit at variable levels (Fig 3).

Interestingly, while no difference in the expression levels of *hsa-miR-nov7* was observed between breast cancer subgroups, we found a significant difference in the expression levels of *has-miR-nov3* related to estrogen receptor status. Thus, the expression levels of *has-miR-nov3* were higher in ER-positive as compared to ER-negative tumors (p = 0.037; Fig 4A). Further,

A. *hsa-miR-nov3*

(i)

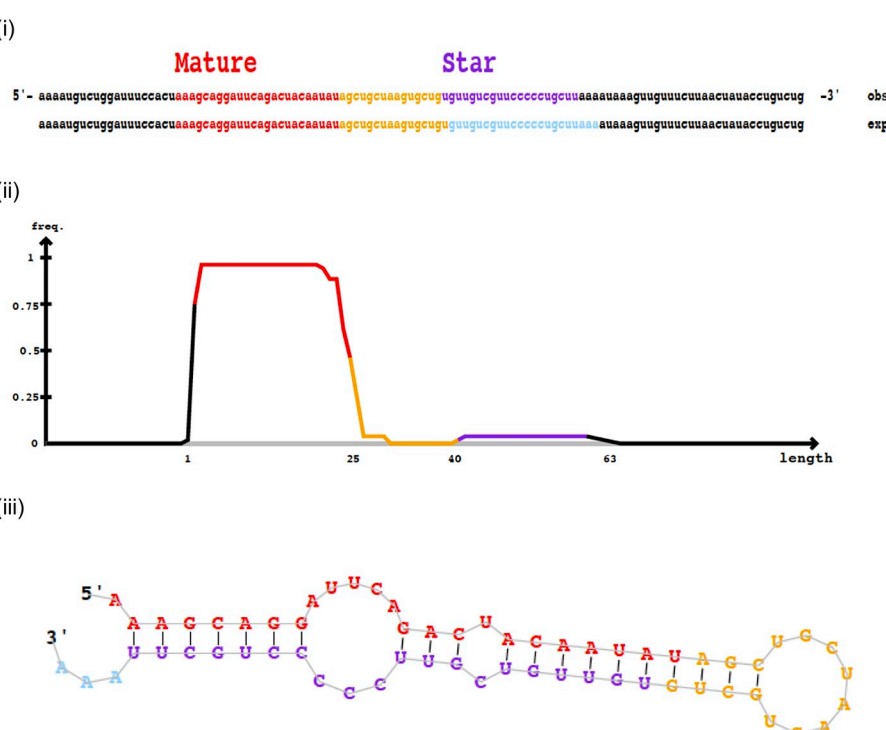

(ii)

(iii)

**Fig 1. Predicted novel miRNAs.** Depiction of novel miRNAs (A) *hsa-miR-nov3* and (B) *hsa-miR-nov7*, identified by miRDeep2, showing (i) predicted mature and star sequences, **exp**, probabilistic model expected from Drosha/Dicer processing and **obs**, observed sequences from sequencing data (ii) density plot for read counts for mature and star sequences as well as (iii) miRNA secondary structure.

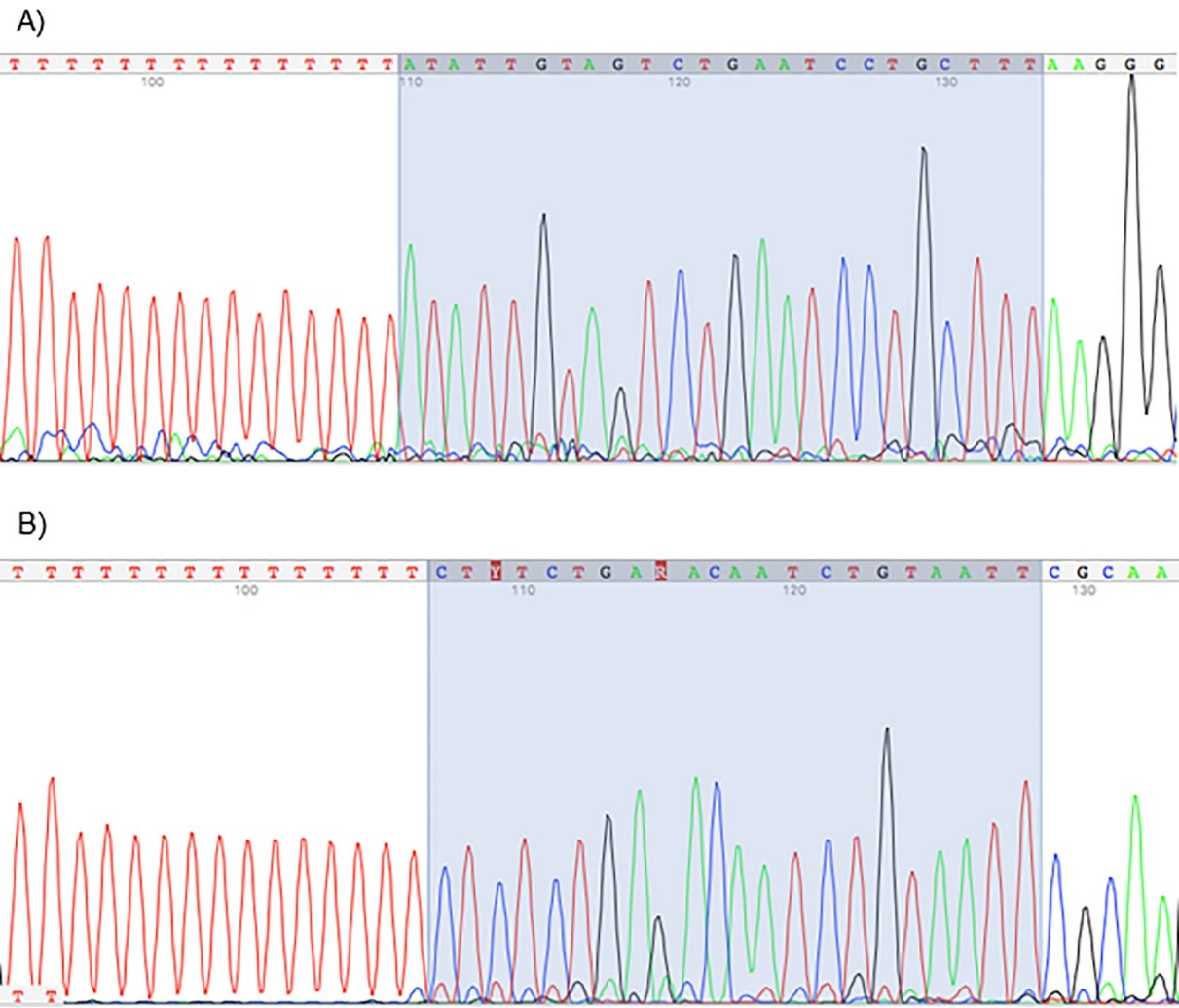

**Fig 2. miRNA sequences.** Chromatogram of capillary-sequenced qPCR products after *hsa-miR-nov3* (A) and *hsa-miR-nov7* (B) amplification. Highlighted background indicates the 22nt miRNA-sequence region (reverse complementary), followed by the Adenine homopolymer indicating *in vitro* adenylation at the expected site, confirming the exact size and sequence of the predicted miRNAs.

assessing the expression levels of the two miRNAs in mRNA-based subclasses of breast cancer according to the Perou classification [51], comparing all five classes, we observed a significant difference between the subtypes with respect to *miR-nov3* expression (p = 0.041; Kruskal-Wallis test; Fig 4C). We found *hsa-miR-nov3* levels to be lower in HER2 like (p = 0.009; Mann-Whitney test) and basal-like (p = 0.04; Mann-Whitney) tumors as compared to tumors of the other classes.

Following the finding that the two miRNAs were detectable in more than 90 percent of patients, in order to assess whether the expression of these miRNAs were tumor specific we compared the levels of *hsa-miR-nov7* and *hsa-miR-nov3* expression in breast cancer tissue versus normal breast tissue. For this purpose, we randomly selected 13 patients from a study where samples of breast tumor tissue and matching normal tissue from a non-tumor bearing quadrant of the same breast were available (study 2, see materials and methods) [28]. We detected expression of the novel miRNAs in both tumor- and normal tissue samples for all 13 patients. Notably, we found *hsa-miR-nov3* expression to be elevated in tumor compared to normal tissue in 10 out of the 13 patients (p = 0.009; Wilcoxon test; Fig 5A). Similar findings

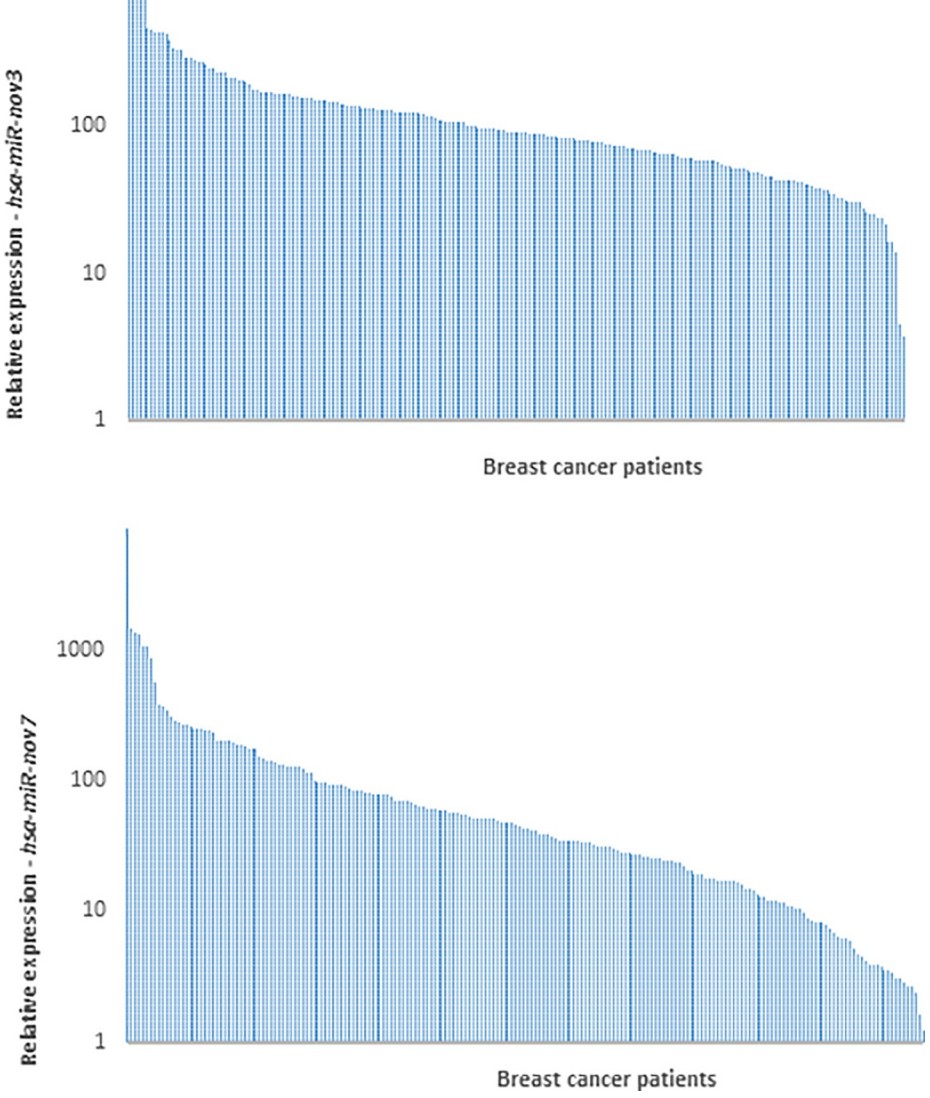

**Fig 3. Expression of novel miRNAs in breast cancer tissue.** Bars indicate the relative expression of *hsa-miR-nov3* (A) and *hsa-miR-nov7* (B) in 223 breast cancer patients.

were observed for *hsa-miR-nov7* with elevated expression in 10 out of 13 tumors (Wilcoxon: p = 0.016; Fig 5B). The level of overexpression (i.e. the ratio of expression levels in tumor versus normal tissue) for the two miRNAs did not correlate to each other (p>0.2; Spearman).

Notably, overexpression of *hsa-miR-nov7* in tumor versus normal tissue was observed predominantly in ER-positive tumors (overexpression in 7 out of 7 ER-positive tumors, contrasting 3 out of 6 ER-negative tumors; p = 0.070; Fischer exact test).

## hsa-miR-nov7 and hsa-miR-nov3 target prediction

Based on our finding of both novel miRNAs to be overexpressed in breast cancer, we next aimed to elucidate the functional roles for *hsa-miR-nov7 and hsa-miR-nov3* by identifying

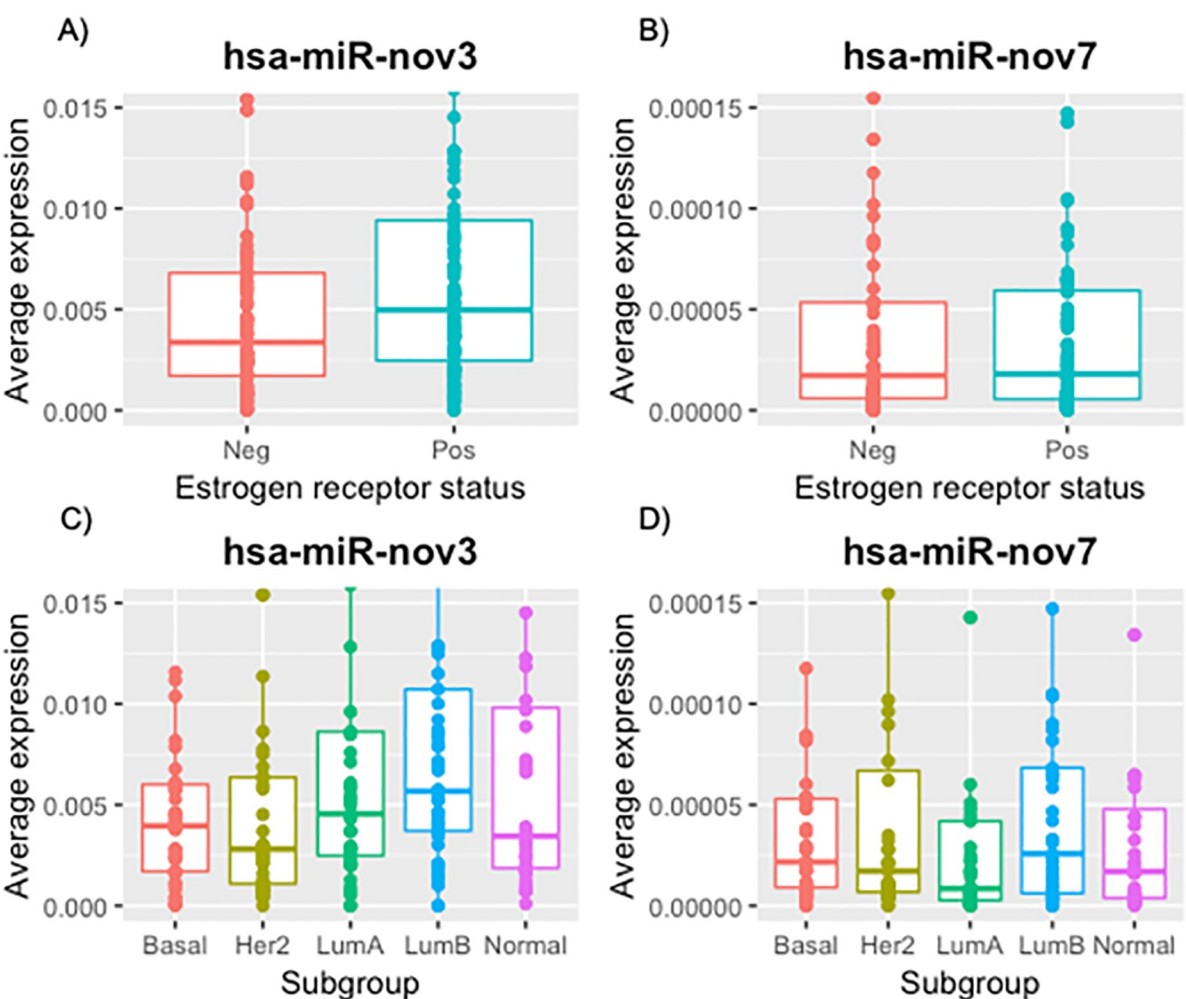

**Fig 4. Expression of novel miRNAs in breast cancer tissue.** Expression levels stratified by ER-status (A, B) and by expression subtypes (C, D).

potential targets. We performed *in silico* target predictions using three different algorithms–miRanda, miRDB and TargetScan Human Custom. miRanda, which predicts possible targets from human transcripts in general, predicted 9200 and 12315 target genes for *hsa-miR-nov7 and hsa-miR-nov3*, respectively. miRDB, which contains curated and possible miRNA targets, predicted 570 and 530 target genes each for *hsa-miR-nov7 and hsa-miR-nov3*, respectively, while TargetScanHuman custom predicted 633 target genes for *hsa-miR-nov7*, and 282 target genes for *hsa-miR-nov3*. For increased stringency in our predictions, we restricted the potential targets to the ones called by all three algorithms (Fig 6). This left a total of 97 and 180 potential targets for *hsa-miR-nov3* and *hsa-mir-nov7*, respectively.

The two lists of 97 and 180 predicted gene targets were then used for KEGG pathway analysis and GO enrichment analysis using GATHER. The top 10 KEGG pathways and GO terms for each microRNAs are listed in Table 2. The KEGG and GO annotations for *hsa-miR-nov3* showed pathways that are important in cell development, communication and cytoskeletal organization. Similar analysis for *hsa-miR-nov7* unveiled pathways playing a vital role in cell functions such as communication and homeostasis. These findings were largely validated by performing the same analyses applying alternative tools (DAVID and topGO; S2 Table).

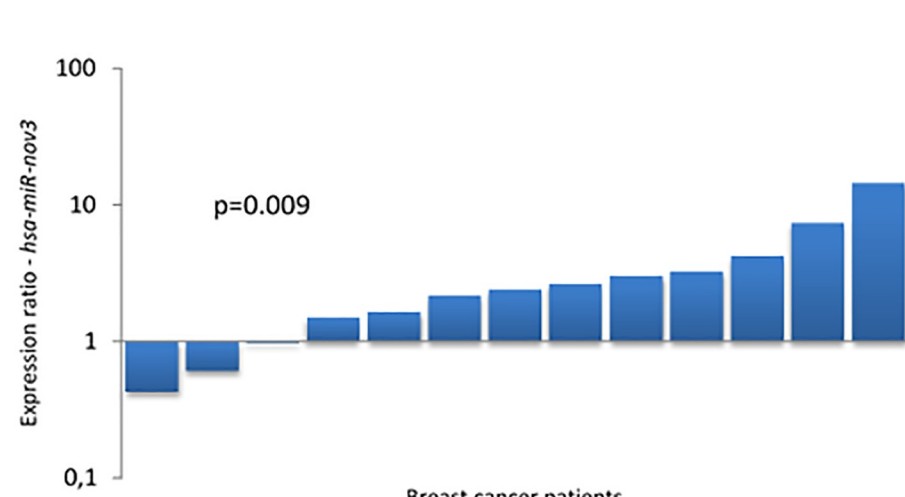

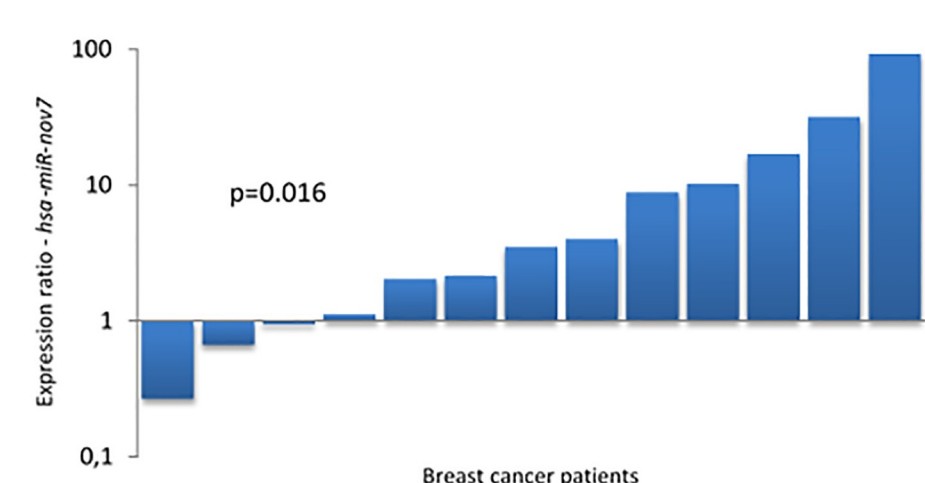

**Fig 5. Expression of novel miRNAs in breast cancer tissue.** Bars indicate the ratio of expression in tumour tissue vs. matched normal breast tissue in 13 breast cancer patients, for *hsa-miR-nov3* (A) and *hsa-miR-nov7* (B).

Thus, both these miRNAs implied cell functions that are vital to cancer development and progression.

In order to further substantiate these *in-silico* predictions, we performed a complete Spearman correlation analysis between the expression levels of *hsa-miR-nov7 and hsa-miR-nov3* and mRNA expression array data available for 203 out of the 233 patients in study 1. Assuming the miRNAs, in general, to execute their function by suppressing gene expression (mRNA degradation), we restricted the analysis to genes which were negatively correlated to expression of the miRNAs. The top ranking negatively correlated genes are listed in Table 3. Notably, the only genes with Rho-values < -0.2 were *RMND5A* for *hsa-miR-nov3* and *GLUD1* and *SASH1* for *hsa-miR-nov7*. Given that the two novel miRNAs were overexpressed in breast cancer tissue, we went on to restrict the correlation analysis to an in-house list of 283-tumor suppressor

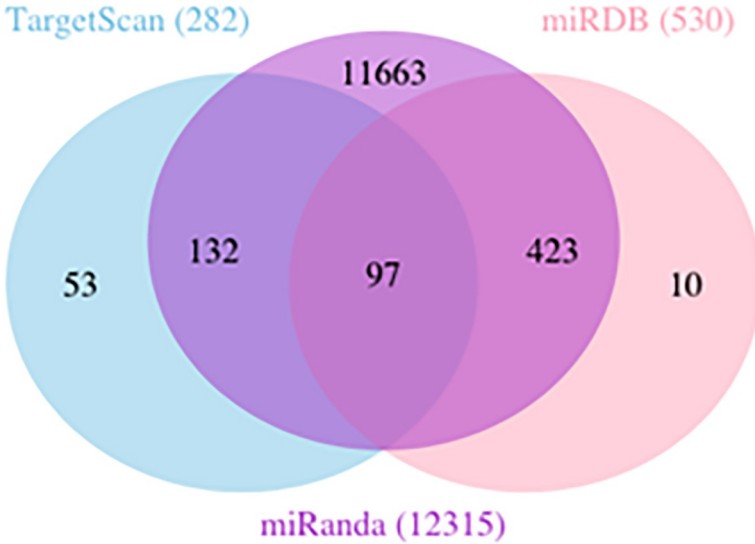

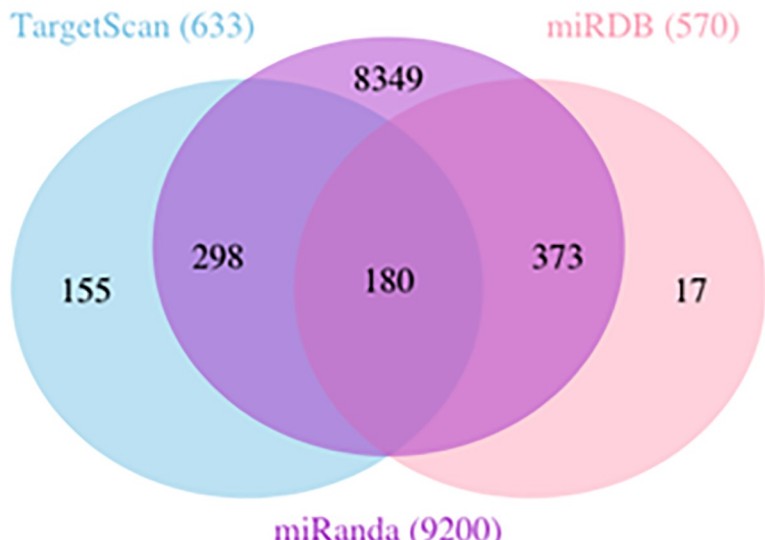

**Fig 6. Target genes predicted.** Venn-diagrams illustrating the number of target genes predicted by TargetScan, mirDB and Miranda for the two novel miRNAs *hsa-mir-nov3* (A) and *hsa-mir-nov7* (B).

**Table 2. Top 10 (arbitrary cut-off) GO and KEGG annotation.**

A) GO annotation—*hsa-miR-nov3*

| # | Annotation | ln(Bayes factor)[a] | neg ln(p value)[b] | FE: neg ln (p value)[c] | FE: neg ln(FDR)[d] |
|---|---|---|---|---|---|
| 1 | GO:0009653 [3]: morphogenesis | 94.88 | 7.98 | 100.5 | 92.91 |
| 2 | GO:0007275 [2]: development | 87.32 | 7.58 | 92.89 | 85.99 |
| 3 | GO:0007154 [3]: cell communication | 85.41 | 7.46 | 90.99 | 84.5 |
| 4 | GO:0009887 [4]: organogenesis | 74.65 | 6.99 | 80.24 | 74.26 |
| 5 | GO:0048513 [3]: organ development | 74.65 | 6.99 | 80.24 | 74.26 |
| 6 | GO:0007165 [4]: signal transduction | 74.2 | 6.97 | 79.77 | 73.97 |
| 7 | GO:0007242 [5]: intracellular signaling cascade | 66.52 | 6.59 | 72.18 | 66.53 |
| 8 | GO:0007010 [6]: cytoskeleton organization and biogenesis | 55.54 | 6.04 | 61.15 | 55.63 |
| 9 | GO:0009790 [3]: embryonic development | 48.63 | 5.7 | 54.28 | 48.88 |
| 10 | GO:0006928 [4]: cell motility | 47.82 | 5.65 | 53.49 | 48.2 |

B) KEGG annotation—*hsa-miR-nov3*

| # | Annotation | Total Genes With Ann | ln(Bayes factor)[a] | neg ln(p value)[b] | FE: neg ln (p value)[c] | FE: neg ln(FDR)[d] |
|---|---|---|---|---|---|---|
| 1 | path:hsa04810: Regulation of actin cytoskeleton | 35 | 9.03 | 4.07 | 13.96 | 9.57 |
| 2 | path:hsa04010: MAPK signaling pathway | 36 | 6.93 | 3.75 | 11.8 | 8.1 |
| 3 | path:hsa04510: Focal adhesion | 32 | 4.15 | 3.26 | 8.94 | 5.94 |
| 4 | path:hsa04110: Cell cycle | 18 | 4.1 | 3.25 | 9.08 | 5.95 |
| 5 | path:hsa04060: Cytokine-cytokine receptor interaction | 33 | 3.23 | 3.07 | 7.97 | 5.24 |
| 6 | path:hsa04620: Toll-like receptor signaling pathway | 17 | 2.97 | 3.01 | 7.92 | 5.24 |
| 7 | path:hsa04210: Apoptosis | 16 | 2.13 | 2.82 | 7.06 | 4.55 |
| 8 | path:hsa04512: ECM-receptor interaction | 14 | 1.17 | 2.55 | 6.09 | 3.72 |
| 9 | path:hsa04630: Jak-STAT signaling pathway | 21 | 1.01 | 2.51 | 5.77 | 3.52 |
| 10 | path:hsa05050: Dentatorubropallidoluysian atrophy (DRPLA) | 5 | 0.7 | 2.41 | 5.9 | 3.59 |

C) GO annotation—*hsa-miR-nov7*

| # | Annotation | ln(Bayes factor)[a] | neg ln(p value)[b] | FE: neg ln(p value)[c] | FE: neg ln(FDR)[d] |
|---|---|---|---|---|---|
| 1 | GO:0007154 [3]: cell communication | 60.17 | 6.3 | 65.79 | 58.14 |
| 2 | GO:0007275 [2]: development | 54.83 | 6 | 60.38 | 53.43 |
| 3 | GO:0007165 [4]: signal transduction | 50.84 | 5.81 | 56.44 | 49.89 |
| 4 | GO:0009653 [3]: morphogenesis | 48.96 | 5.72 | 54.56 | 48.3 |

(*Continued*)

**Table 2.** (Continued)

| 5 | GO:0050794 [3]: regulation of cellular process | 41.31 | 5.3 | 46.94 | 40.9 |
|---|---|---|---|---|---|
| 6 | GO:0009987 [2]: cellular process | 40.56 | 5.26 | 46.33 | 40.48 |
| 7 | GO:0009887 [4]: organogenesis | 40.37 | 5.25 | 45.94 | 40.24 |
| 8 | GO:0048513 [3]: organ development | 39.98 | 5.23 | 45.54 | 40.09 |
| 9 | GO:0007242 [5]: intracellular signaling cascade | 39.87 | 5.22 | 45.58 | 40.09 |
| 10 | GO:0050789 [2]: regulation of biological process | 39.18 | 5.18 | 44.62 | 39.27 |

D) KEGG annotation—*hsa-miR-nov7*

| # | Annotation | Total Genes With Ann | ln(Bayes factor)[a] | neg ln(p value)[b] | FE: neg ln(p value)[c] | FE: neg ln(FDR)[d] |
|---|---|---|---|---|---|---|
| 1 | path:hsa04630: Jak-STAT signaling pathway | 27 | 5.53 | 3.5 | 10.48 | 6.6 |
| 2 | path:hsa04350: TGF-beta signaling pathway | 18 | 5.22 | 3.45 | 10.3 | 6.6 |
| 3 | path:hsa04010: MAPK signaling pathway | 33 | 3.15 | 3.04 | 7.94 | 4.57 |
| 4 | path:hsa04210: Apoptosis | 17 | 2.51 | 2.91 | 7.49 | 4.27 |
| 5 | path:hsa04620: Toll-like receptor signaling pathway | 17 | 2.28 | 2.85 | 7.25 | 4.24 |
| 6 | path:hsa04020: Calcium signaling pathway | 4 | 2.23 | 2.84 | 0 | 0 |
| 7 | path:hsa00471: D-Glutamine and D-glutamate metabolism | 3 | 1.12 | 2.54 | 6.48 | 3.7 |
| 8 | path:hsa04510: Focal adhesion | 29 | 0.96 | 2.49 | 5.64 | 3.23 |
| 9 | path:hsa05030: Amyotrophic lateral sclerosis (ALS) | 5 | -0.17 | 0 | 5.04 | 2.78 |
| 10 | path:hsa04512: ECM-receptor interaction | 13 | -0.28 | 0 | 4.61 | 2.39 |

[a] Measure of the strength of annotation

[b] p-value for the Bayes factor estimate

[c] p-value for Fishcer's exact test

[d] FDR for Fishcer's exact test

**Table 3. Spearman correlation table for *hsa-miR-nov3* and *hsa-miR-nov7* and their top 25 target genes (arbitrary cut-off for inclusion in the table; ranked by inverse correlation).**

A) *hsa-miR-nov3*

| Gene Symbol | Estimate | P.value | Expression (mean) |
|---|---|---|---|
| RMND5A | -0.2018 | 0.0038 | 14.0750 |
| YES1 | -0.1649 | 0.0184 | 17.0218 |
| PALM2-AKAP2 | -0.1455 | 0.0378 | 13.0997 |
| SLC7A1 | -0.1224 | 0.0811 | 16.8650 |
| RAPGEF5 | -0.1208 | 0.0853 | 14.9652 |
| CTDSPL2 | -0.1196 | 0.0885 | 15.4945 |
| SLC4A5 | -0.1077 | 0.1251 | 15.0101 |
| HIPK1 | -0.1046 | 0.1366 | 13.3737 |
| ABHD12 | -0.0998 | 0.1555 | 16.2313 |
| FMNL2 | -0.0982 | 0.1624 | 16.0939 |
| POU4F1 | -0.0933 | 0.1844 | 13.4684 |
| RPS6KA3 | -0.0905 | 0.1981 | 14.6430 |
| LARP1 | -0.0890 | 0.2054 | 15.0210 |
| WIPI2 | -0.0702 | 0.3184 | 14.7316 |
| MTCH1 | -0.0575 | 0.4139 | 18.6604 |
| DIAPH1 | -0.0528 | 0.4530 | 16.7109 |
| MARCKS | -0.0481 | 0.4946 | 18.6286 |
| LUZP1 | -0.0453 | 0.5200 | 17.1097 |
| DNAJC8 | -0.0449 | 0.5238 | 18.2152 |
| CLOCK | -0.0436 | 0.5354 | 15.7894 |
| SLAMF6 | -0.0415 | 0.5557 | 15.4277 |
| CDAN1 | -0.0405 | 0.5655 | 16.6394 |
| PCDH11X | -0.0359 | 0.6104 | 13.4661 |
| RYBP | -0.0346 | 0.6234 | 16.9184 |
| FGF1 | -0.0344 | 0.6249 | 13.9423 |

B) *hsa-miR-nov7*

| Gene Symbol | Estimate | P.value | Expression (Mean) |
|---|---|---|---|
| GLUD1 | -0.2274 | 0.0011 | 18.0399 |
| SASH1 | -0.2095 | 0.0026 | 16.9164 |
| MARK1 | -0.1883 | 0.0070 | 15.0356 |
| ARID5B | -0.1877 | 0.0072 | 17.7569 |
| ELOVL5 | -0.1854 | 0.0079 | 17.5656 |
| PUM1 | -0.1707 | 0.0147 | 17.8295 |
| PNRC2 | -0.1599 | 0.0224 | 15.4674 |
| UNC13B | -0.1583 | 0.0238 | 15.5633 |
| FLRT2 | -0.1581 | 0.0239 | 15.7323 |
| ZFHX4 | -0.1482 | 0.0344 | 14.7383 |
| CHIC1 | -0.1479 | 0.0348 | 13.5807 |
| MAN1A1 | -0.1457 | 0.0375 | 15.4956 |
| CPEB2 | -0.1387 | 0.0478 | 14.6995 |
| PDE4D | -0.1377 | 0.0495 | 13.9823 |
| TMED7 | -0.1366 | 0.0514 | 17.1083 |
| NDFIP1 | -0.1280 | 0.0680 | 16.1458 |
| CSMD1 | -0.1269 | 0.0704 | 13.8158 |
| MITF | -0.1187 | 0.0908 | 14.0482 |

(*Continued*)

**Table 3.** (Continued)

| | | | |
|---|---|---|---|
| ITSN1 | -0.1185 | 0.0915 | 14.8011 |
| CTDSPL2 | -0.1178 | 0.0932 | 15.4945 |
| ATAD2B | -0.1178 | 0.0932 | 14.9892 |
| SFRP2 | -0.1129 | 0.1080 | 18.4511 |
| DPP10 | -0.1119 | 0.1110 | 13.4306 |
| BMPR2 | -0.1107 | 0.1149 | 17.1664 |
| EIF5A2 | -0.1100 | 0.1174 | 14.5450 |

genes previously described. Among these tumor suppressors, we found 115 to be negatively correlated to *hsa-miR-nov7* and 119 to *hsa-miR-nov3* (S3 Table). Assessing the intersection between these negatively correlated tumor suppressor genes and the predicted targets, we obtained a list of one gene for *hsa-miR-nov3* (*ATRX*) and three genes for *hsa-miR-nov7* (*APC*, *SFRP2* and *CDH11*), but the correlations were non-significant in all 4 cases (Table 4, Fig 7).

In order to get a broader overview of potential biological function, we selected the 100 gene transcripts with the strongest positive and the top 100 gene transcripts with the strongest negative correlation to the two miRNAs (independent of previous target-predictions) and performed gene ontology analyses. We detected no cancer related pathways or cellular functions to be significantly associated with *hsa-miR-nov7* (S4 Table). However, for *hsa-miR-nov3*, KEGG analysis of the negatively correlated genes revealed associations to Hepatorcellular carcinoma as well as several pathways related to drug metabolism (S5 Table). Notably, when seeking to validate these findings by application of alternative tools (DAVID and topGO) the latter was not validated. (S6 and S7 Tables).

## Expression of hsa-miR-nov7 and hsa-miR-nov3 and clinical outcome in breast cancer

Since both *hsa-miR-nov7* and *hsa-miR-nov3* were overexpressed in the tumor tissue of breast cancer patients, we assessed whether any of the two novel miRNAs were associated to clinical outcomes in study 1 (223 breast cancer patients). Given that these patients were enrolled in a prospective study specifically designed to assess response to primary chemotherapy administered as epirubicin or paclitaxel monotherapy in a neoadjuvant setting [27, 52], we assessed the association of *hsa-miR-nov7 and hsa-miR-nov3* levels with primary therapy response and with long term survival (10-years).

We found no association between any of the two novel miRNAs and primary response to either epirubicin or paclitaxel (S8 Table). Regarding survival, we observed a weak association between high levels of *hsa-miR-nov7* and poor survival in the paclitaxel treated arm of the study, with the strongest associations observed for relapse free survival, however, none of these associations reached statistical significance (Fig 8). No effect was observed in the epirubicin treated arm. Further, for *hsa-miR-nov3*, no significant correlation to outcome was recorded.

**Table 4. List of intersection between correlated tumour suppressor genes and the predicted targets of *hsa-miR-nov3* and *hsa-miR-nov7*.**

| *hsa-miR-nov3* | *hsa-miR-nov7* |
|---|---|
| ATRX | APC |
| | CDH11 |
| | SFRP2 |

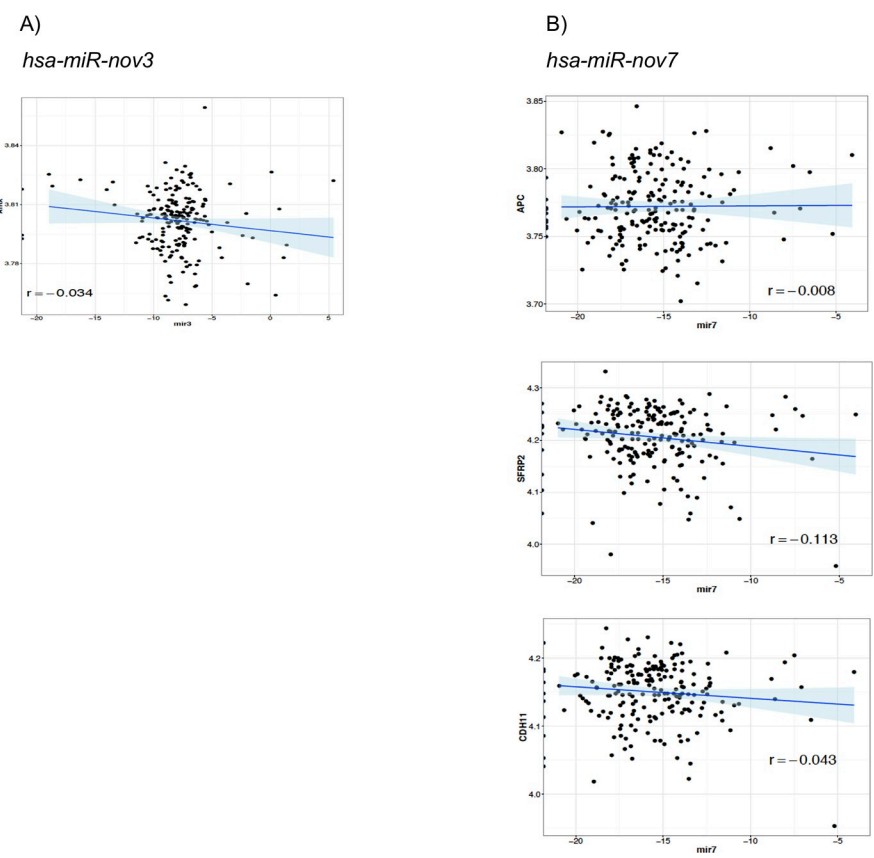

**Fig 7. Correlations to tumor suppressor genes.** Scatter plots showing correlation of target tumor suppressors with A) *hsa-miR-nov3* and B) *hsa-miR-nov7*.

Given the skewed expression levels between breast cancer subtypes for *hsa-miR-nov3*, we performed survival analyses stratified for ER-status and subtypes. These analyses revealed no significant associations to survival (Log rank test p-values ranging from 0.09 to 0.98).

## Discussion

We investigated whether we could detect novel, previously undescribed miRNAs and, if so, address their potential association to other defined biological parameters and to outcome in a cohort of locally advanced breast cancer. We successfully predicted 10 new miRNAs, out of which 2 were deemed reliable because of their detected presence in more than one patient. Although these two novel miRNAs (preliminary termed *hsa-miR-nov7* and *hsa-miR-nov3*) were only predicted from 8 samples among the 50 initially sequenced biopsies, we found them to be expressed in all patients by highly sensitive qPCR at varying levels. In addition to our *in vitro* validations, the qPCR detection validated the initial NGS based analysis, detecting these two miRNAs.

Since expression of the two miRNAs was confirmed in breast tumor tissue from the majority of patients analyzed, we went on to assess the relative expression levels in tumor versus matched normal breast tissue, collected from a non-tumor bearing quadrant. Our finding that both novel miRNAs had higher expression levels in tumor than in normal tissue indicates a potential functional role in breast cancer. However, although being overexpressed, the biological role of these two miRNAs in cancer should be interpreted with caution. The expression

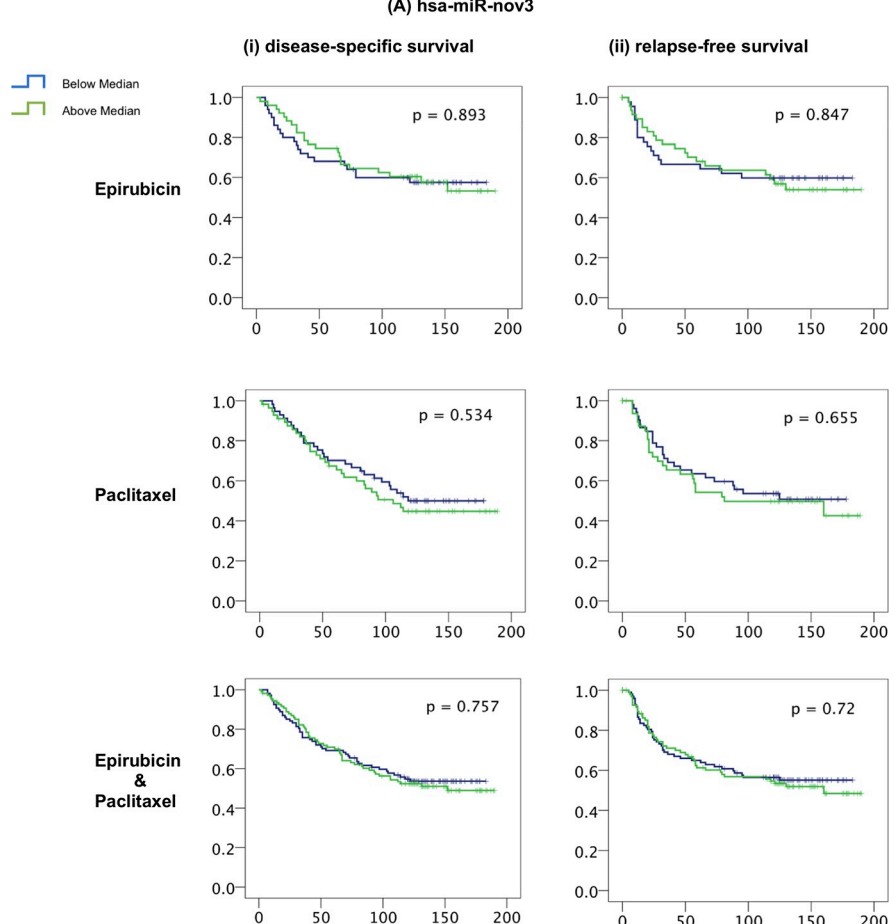

**Fig 8. miRNAs and breast cancer survival.** Kaplan-Meier curves showing (i) disease-specific and (ii) relapse-free survival of locally advanced breast cancer patients treated with epirubicin or paclitaxel monotherapy in the neoadjuvant setting (study 1), with respect to expression levels of (A) hsa-miR-nov3 and (B) hsa-miR-nov7 on all samples.

levels are very low, and it is therefore uncertain whether they will have a major impact on cellular functions. Notably, given our approach and identification of the two miRNAs with low expression level, this indicates that there may currently be a limited potential for new discoveries of miRNAs high expression levels and strong functional roles in breast cancer. However, when assessing the potential functional roles of these microRNAs by *in silico* prediction of targets followed by validation using correlation to mRNA-array data, the KEGG and GO annotations for these targets revealed cellular functions of potential importance in development and progression of cancer. As such, our present findings may warrant further investigations into the functions of the two miRNAs. Notably, regarding *hsa-miR-nov3*, it was of particular interest that this miRNA was significantly higher expressed in ER-positive as compared to ER-negative breast cancers. Accordingly, we found relatively high expression levels of *hsa-miR-nov3* in tumors of the luminal and normal-like subtypes, contrasting low expression levels in basal-like and her2-like tumors [53, 54]. This finding may indicate a potential role for *hsa-miR-nov3* restricted to ER-positive tumors.

Regarding potential specific targets, we narrowed these down by first assessing the intersect of three different target prediction algorithms, and then the intersect of this result with a

predefined list of tumor suppressors. Although none of the remaining genes after this filtering had a statistically significant inverse correlation with the miRNAs, we identified some potentially interesting connections: For *hsa-miR-nov3*, we propose *ATRX* as a target. This is a gene in the SWI/SNF family, involved in chromatin remodelling, and it has previously been found subject to loss of heterozygosity (LOH) in breast cancer [55]. Importantly, we recently reported mutations in the SWI/SNF family genes to be enriched in relapsed breast cancer as compared to primary cancers [56]. Thus, this supports the hypothesis of a breast cancer promoting function for *hsa-miR-nov3*. For *hsa-miR-nov7*, we propose *APC*, *SFRP2*, and *CDH11* as potential targets. Interestingly, the two former are involved in regulation of the Wnt-signalling pathway [57–59] and both have previously been reported as targets for several miRNAs in breast cancer [60–62]. Taken together, this may imply a role for *hsa-miR-nov7* in Wnt signaling. Notably, during our work with the present project, *hsa-miR-nov7*, was identified by Lim and colleagues and coined miR-10393-3p [50]. They found this miRNA to target genes involved in chromatin modifications associated with pathogenesis of Diffuse large B-cell lymphoma (DLBCL). While this differs from our present finding, it may likely be explained by tissue specific effects of the miRNA.

Regarding any predictive or prognostic role for the two investigated miRNAs, we found no significant impact on survival. While we recorded a non-significant trend towards an association between miRnov7 expression and overall survival in the paclitaxel arm, further studies on larger patient cohorts are warranted to clarify this issue. Alternatively, the miRNAs could play a role in tumorigenesis but not later tumor progression. As such, the observed overexpression in tumor tissue compared to normal breast tissue may be a remaining signal from tumorigenesis.

Whether cancer related overexpression of the two miRNAs described here is merely consequences of other molecular mechanisms in cancer cells or whether the two miRNAs may be involved in tumorigenesis, but not subsequent cancer progression, remains unknown.

## Supporting information

**S1 Fig. Predicted novel miRNAs.** Table on the upper left shows miRDeep2 scores and read counts. RNA secondary structure for miRNA on the top right. Color code for depiction as follows mature sequence in red, loop sequence in yellow and purple for star sequences. Density plot in the middle shows distribution of reads in precursor reads predicted. Dotted lines illustrate alignment and mm, number of mismatches. Exp, is potential precursor model predicted by algorithm with taking accounts of stability based on free energy, position and read frequencies according to Dicer/Drosha processing of miRNA. Obs, is position and reads found from deep sequencing data. (A) *hsa-miR-nov3* and (B) *hsa-miR-nov7*.
(DOCX)

**S1 Table. In-house pan-cancer panel of 283 tumor suppressor genes.** Panel generated based on CGPv2/3-panels [41], Roche's Comprehensive Cancer Design along with manual literature search, to filter target genes of interest.
(DOCX)

**S2 Table. Predicted miRNA-targets by DAVID and topGO.**
(XLSX)

**S3 Table. Correlation miRNAs and tumour suppressor genes.** Spearman correlation table for *hsa-miR-nov3* (A) and *hsa-miR-nov7* (B) inversely correlated tumor suppressor genes.
(DOCX)

**S4 Table. Correlations mir7 and gene ontology.**
(XLS)

**S5 Table. Correlations mir3 and gene ontology.**
(XLS)

**S6 Table. Negatively correlated genes (validation analyses).**
(XLSX)

**S7 Table. Positively correlated genes (validation analyses).**
(XLSX)

**S8 Table. Statistics mir3 and mir7 versus response to treatment.**
(XLSX)

**S1 File. Supporting information mirDeep.**
(DOCX)

## Acknowledgments

We thank Beryl Leirvaag and Gjertrud T. Iversen for technical assistance.

## Author Contributions

**Conceptualization:** Deepak Poduval, Stian Knappskog.

**Data curation:** Deepak Poduval, Zuzana Sichmanova, Stian Knappskog.

**Formal analysis:** Deepak Poduval, Zuzana Sichmanova, Anne Hege Straume, Stian Knappskog.

**Funding acquisition:** Per Eystein Lønning, Stian Knappskog.

**Methodology:** Anne Hege Straume, Stian Knappskog.

**Supervision:** Per Eystein Lønning, Stian Knappskog.

**Writing – original draft:** Deepak Poduval, Stian Knappskog.

**Writing – review & editing:** Deepak Poduval, Per Eystein Lønning, Stian Knappskog.

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
