## [Decision Letter · Decision Letter 0]

7 Jan 2020

PONE-D-19-30537

The novel microRNAs hsa-miR-nov7 and hsa-miR-nov3 are over-expressed in locally advanced breast cancer

PLOS ONE

Dear Dr Knappskog,

Thank you for submitting your manuscript to PLoS ONE. Your manuscript has been reviewed by two experts in the field and their comments are appended below. After reading the reviews and looking at the manuscript, we feel that your study has merit, but is not suitable for publication as it currently stands. Therefore, my decision is "Major Revision”.

You must revise accordingly and explain your revisions in a covering letter if you wish for us to consider your paper further for publication. Note that it will have to go through another round of review.

We invite you to submit a revised version of the manuscript that addresses the concerns raised by the reviewers and myself. Please pay attention to all the suggestions and give them due consideration.

Specifically:

You should answer to the comments of Reviewer 1. As raised by the reviewer, NGS technique can give several biases and validation should include Northern blot, as frequently recommended (see Alles et al. Nucleic Acids Research, 2019, Vol. 47, No. 7 3353–3364). It would be also highly recommended to add additional functional data to explore the potential relevance of these miRNAs, as suggested by the reviewer.

You should also clearly respond to the three major concerns raised by Reviewer 2. Several methodological and statistical points have to be clarified and / or completed.

Finally, I have also personally raised two additional points: first, the authors should submit the whole small RNA-Seq dataset to a public database such as Gene Expression Omnibus (https://www.ncbi.nlm.nih.gov/geo/). Second, they should also match (and eventually submit) their prediction with a central repository for collecting miRNA candidates such as miRCarta (Backes et al. Nucleic Acids Res. 2018 Jan 4;46(D1):D160-D167) to check for potential redundancies within their sequences.

We would appreciate receiving your revised manuscript by Feb 21 2020 11:59PM. To enhance the reproducibility of your results, we recommend that if applicable you deposit your laboratory protocols in protocols.io, where a protocol can be assigned its own identifier (DOI) such that it can be cited independently in the future. For instructions see: http://journals.plos.org/plosone/s/submission-guidelines#loc-laboratory-protocols

We look forward to receiving your revised manuscript.

Kind regards,

Bernard Mari, Ph.D

Academic Editor

PLOS ONE

Journal Requirements:

3. Thank you for including your ethics statement: 

"The study involves use of human material from cancer patients. All patients provided written informed consent, and the studies conducted in accordance to national laws, regulation and ethical permissions."

Reviewers' comments:

Reviewer's Responses to Questions

**Comments to the Author**

1. Is the manuscript technically sound, and do the data support the conclusions?

Reviewer #1: Yes

Reviewer #2: Yes

2. Has the statistical analysis been performed appropriately and rigorously? 

Reviewer #1: Yes

Reviewer #2: Yes

3. Have the authors made all data underlying the findings in their manuscript fully available?

Reviewer #1: Yes

Reviewer #2: Yes

4. Is the manuscript presented in an intelligible fashion and written in standard English?

Reviewer #1: Yes

Reviewer #2: Yes

5. Review Comments to the Author

Reviewer #1: The authors described the over-expression of two novel miRNAs, hsa-miRNA-nov3 and hsa-miRNA-nov7 in breast cancer. Out of these, one was not reported previously and the other not reported previously in breast cancer. The in-silico analysis to explore the putative functional role is well designed and follows a logical and systematic approach. However, there is no strong experimental evidence of their relevance to biological function, probably because of their low expression levels, as mentioned by authors. Moreover, miRNA nov3 has statistically insignificant association with patient survival, while miR-nov7 has no association with patient survival. Despite the computational approach to explore the potential functional relevance of these miRNAs, the significance of this work in the larger context is weak, provided lack of association between miRNAs and patients’ survival. Moreover, based on the evidence presented, it is hard to distinguish whether miRNAs are involved in tumorigenesis or their over-expression is a mere co-incidental effect of some other molecular pathway.

Also, NGS technique is well used to study miRNAs but it is not a strong system to find novel miRNAs. It is more accurate to use Northern blot to claim novel miRNAs.

Reviewer #2: In this manuscript, the authors have analyzed small RNA sequencing data for 50 locally advanced breast cancers to identify novel miRNAs. 10 novel miRNAs were predicted using mirDeep software and found 2 miRNAs (hsa-miR-nov3 & hsa-miR-nov7) over-expressed in tumor versus normal breast tissue in a separate set of 13 patients. They found hsa-miR-nov3 expressed at higher levels in ER-positive as compared to ER-negative tumors. They also predicted target genes for these 2 microRNAs and identified inversely correlated genes in mRNA expression array data available from 203 out of the 223 patients. Also, they have done KEGG and GO annotations to target genes that revealed pathways essential to cell development, communication and homeostasis. Likewise, they observed a weak association between high expression levels of hsa-miR-nov7 and poor survivals that did not reach statistical significance. As the author has done a very systemic work, but before publication, following issues should be addressed:

1. The author has written the introduction section that describes the role of mirna in cancer. Author has given examples of mirna that are involved in breast cancer. It would be very useful if the author provides some latest study for mirna that have been used as a biomarker in cancer diagnosis.

2. As there are number of gene enrichment tools (like DAVID, GSEA). Authors may also use other tool also for functional gene enrichment.

3. As given in Table 2 and 3, what are the criteria for selecting top 10 GO annotation as well as KEGG annotation. Also, please define how Bayes factor, p-value and FDR value calculated?

6. PLOS authors have the option to publish the peer review history of their article (what does this mean?). If published, this will include your full peer review and any attached files.

Reviewer #1: No

Reviewer #2: Yes: Manoj Kumar

---

## [Author Response · Author response to Decision Letter 0]

20 Feb 2020

Response to Editor and Reviewers regarding manuscript PONE-D-19-30537

“The novel microRNAs hsa-miR-nov7 and hsa-miR-nov3 are over-expressed in locally advanced breast cancer”

In the following, comments from the Editor and Reviewers are given in blue italic, while our responses are given in normal black font.

Comments from the Editor

Specifically:

You should answer to the comments of Reviewer 1. As raised by the reviewer, NGS technique can give several biases and validation should include Northern blot, as frequently recommended (see Alles et al. Nucleic Acids Research, 2019, Vol. 47, No. 7 3353–3364). It would be also highly recommended to add additional functional data to explore the potential relevance of these miRNAs, as suggested by the reviewer.

You should also clearly respond to the three major concerns raised by Reviewer 2. Several methodological and statistical points have to be clarified and / or completed.

Response:

Please see our responses to these issues below, under our responses to the Reviewers’ specific comments.

Finally, I have also personally raised two additional points: first, the authors should submit the whole small RNA-Seq dataset to a public database such as Gene Expression Omnibus (https://www.ncbi.nlm.nih.gov/geo/). 

Response:

We fully concur with this and have now submitted the dataset as requested. The accession number is GSE145151. This information has now been included in the Materials and Methods section of the revised manuscript. 

Second, they should also match (and eventually submit) their prediction with a central repository for collecting miRNA candidates such as miRCarta (Backes et al. Nucleic Acids Res. 2018 Jan 4;46(D1):D160-D167) to check for potential redundancies within their sequences.

Response:

We agree with the Reviewer and have now cross checked the potential presence of our novel miRNAs against human precursor miRNA sequences obtained from miRCarta database (Backes et al., 2018) (curated from miRNA information from databases such as miRbase, miRmaster, TargetScan). Alignment of our sequences against the miRCarta database resulted in no hits for any of our two miRNAs presented in the present manuscript, reinforcing the novelty of the miRNAs.

This cross check and its result is now included in the Results section of the revised manuscript (page 7).

Journal Requirements:

1. Please ensure that your manuscript meets PLOS ONE's style requirements, including those for file naming. The PLOS ONE style templates can be found at http://www.plosone.org/attachments/PLOSOne_formatting_sample_main_body.pdf and 

http://www.plosone.org/attachments/PLOSOne_formatting_sample_title_authors_affiliations.pdf

Response:

This has now been checked for the revised manuscript.

Response:

We have now removed the phrase “data not shown” with respect to the assessment of potential correlation between miRNA expression levels and response to therapy. Instead, we now refer to a new Supporting Table (Supporting Table S5) where we specify the statistics performed and the resulting p-values, as well as give the full data set behind the calculations. We have also removed the phrase “data not shown” with respect to survival analyses in subgroups breast cancers. Instead, we now state the range of the non-significant p-values for these analyses. 

3. Thank you for including your ethics statement: 

"The study involves use of human material from cancer patients. All patients provided written informed consent, and the studies conducted in accordance to national laws, regulation and ethical permissions."

Response:

We have now included the full name of the REC approving our study and biobanks in the Material and Method section, page 4.

Response:

We have now revised the submission form accordingly.

Reviewers' comments:

Reviewer's Responses to Questions

Comments to the Author

1. Is the manuscript technically sound, and do the data support the conclusions?

Reviewer #1: Yes

Reviewer #2: Yes

2. Has the statistical analysis been performed appropriately and rigorously? 

Reviewer #1: Yes

Reviewer #2: Yes

3. Have the authors made all data underlying the findings in their manuscript fully available?

Reviewer #1: Yes

Reviewer #2: Yes

4. Is the manuscript presented in an intelligible fashion and written in standard English?

Reviewer #1: Yes

Reviewer #2: Yes

5. Review Comments to the Author

Reviewer #1: 

The authors described the over-expression of two novel miRNAs, I-miRNA-nov3 and I-miRNA-nov7 in breast cancer. Out of these, one was not reported previously and the other not reported previously in breast cancer. The in-silico analysis to explore the putative functional role is well designed and follows a logical and systematic approach. However, there is no strong experimental evidence of their relevance to biological function, probably because of their low expression levels, as mentioned by authors. Moreover, miRNA nov3 has statistically insignificant association with patient survival, while miR-nov7 has no association with patient survival. Despite the computational approach to explore the potential functional relevance of these miRNAs, the significance of this work in the larger context is weak, provided lack of association between miRNAs and patients’ survival. Moreover, based on the evidence presented, it is hard to distinguish whether miRNAs are involved in tumorigenesis or their over-expression is a mere co-incidental effect of some other molecular pathway.

Response:

We understand these concerns from the reviewer and to a large extent we agree. 

Regarding functional role of the two miRNAs, we have now performed additional computational analyses using different tools in order to validate our original findings. This is further described in our response to Reviewer #2’s point 2. 

Regarding the functional relevance / biological importance of the two miRNAs, we discussed (as pointed out by the reviewer) that the functions may be weak due to low expression levels. The fact that we performed our open screen and detected these two miRNAs (but not others in more than single patients), we believe to be an important message per se; it gives a message to the scientific community that the likelihood of detecting novel miRNAs in breast cancer with high expression levels and important functions in a majority of patients, may be limited. We have now emphasized this point in the Discussion section of the revised manuscript.

Also, NGS technique is well used to study miRNAs but it is not a strong system to find novel miRNAs. It is more accurate to use Northern blot to claim novel miRNAs.

Response:

This point is well taken. We fully agree that NGS is not an optimal strategy for identification of novel miRNAs on its own, but that it needs subsequent validation. In the present work, we used NGS as an initial screen for potential novel miRNAs, with an open approach, and then moved on to validation. Regarding Northern blot analyses, we agree that this could be one way of validating the presence of miRNAs. The reason for this is two-fold: Northern will give some information about sequence (by binding of a probe with known sequence) and a size estimate (by migration under electrophoresis). In the present study, we have used a validation set-up that we believe is superior to Northern with respect to both these measures: We have poly-A-tagged, amplified, cloned and sequenced the patient’s miRNA. This has enabled us to determine both the exact sequence and the exact size of the novel miRNAs (results shown in Figure 2). We believe this is the best possible validation of their presence and nature. However, when reading the comments from the Reviewer and the Editor, we realize that this point was not properly explained in the original manuscript and have now revised the text to make this more clear for the reader: We have added a new paragraph to the Materials and Methods section (page 5) as well as added sentences specifying this point both in the main text of the Results section and the legend to Figure 2 (page 8).

Reviewer #2:

In this manuscript, the authors have analysed small RNA sequencing data for 50 locally advanced breast cancers to identify novel miRNAs. 10 novel miRNAs were predicted using mirDeep software and found 2 miRNAs (I-miR-nov3 & I-miR-nov7) over-expressed in tumor versus normal breast tissue in a separate set of 13 patients. They found I-miR-nov3 expressed at higher levels in ER-positive as compared to ER-negative tumors. They also predicted target genes for these 2 microRNAs and identified inversely correlated genes in mRNA expression array data available from 203 out of the 223 patients. Also, they have done KEGG and GO annotations to target genes that revealed pathways essential to cell development, communication and homeostasis. Likewise, they observed a weak association between high expression levels of I-miR-nov7 and poor survivals that did not reach statistical significance. As the author has done a very systemic work, but before publication, following issues should be addressed:

1. The author has written the introduction section that describes the role of mirna in cancer. Author has given examples of mirna that are involved in breast cancer. It would be very useful if the author provides some latest study for mirna that have been used as a biomarker in cancer diagnosis.

Response:

This point I well taken. We feel the introduction was well balanced, so we have not added very much new text, but in the revised manuscript, we have now included information about the use of miRNAs a biomarkers for cancer diagnosis, as suggested by the Reviewer, and refer to a recent review with detailed overview of the original literature in the field.

2. As there are number of gene enrichment tools (like DAVID, GSEA). Authors may also use other tool also for functional gene enrichment.

Response:

We fully agree with this and realize we could reach further validations regarding the potential biological roles of the miRNAs by applying additional gene enrichment tools. In the revised manuscript, we have included analyses both by DAVID and topGO. These are now included as validations in the Results section and we have given the detailed output as the new Supporting Table S2. (Note that this new table has caused re-numbering of other Supporting tables). 

3. As given in Table 2 and 3, what are the criteria for selecting top 10 GO annotation as well as KEGG annotation. Also, please define how Bayes factor, p-value and FDR value calculated?

Response:

The selection of the top 10 annotations (Table 2) and the top 25 correlations (Table 3) were based on an arbitrary cut-offs, simply for reader friendliness of the Tables. We realize that this was not clear, and have now modified the legends to make this clearer for the reader.

 Bayes factor, p-values and FDR were all calculated using the GATHER-algorithm (Chang and Nevins, 2006). 

Here, Bayes factor is a measure of the strength of annotation. The actual calculations and the formulas are given in the original publication (Chang and Nevins, 2006); in brief, the comparison is based on a 2x2 table where one dimension is annotations associated with 2 compared groups and the other dimension is the number of genes associated with the annotation. In the output (shown in our table 2), Bayes factor is given in one column, with a p-value related to the Bayes factor estimate in the next column.

In addition, the 2x2 comparison can be assessed by a regular Fischer’s exact test. This is given as “FE” in the output, along with the FDR based on this Fischer’s exact test. Both p-values and the FDR are given as ln of the actual values as default (for a more compact representation). 

Note that the precision of the method was also assessed in detail in the original publication (please see figure 2b in the original publication (Chang and Nevins, 2006))

We realize that these points were not evident from the presentation in our original manuscript and that the headings in Table 2 were not explained to the reader. We have now clarified this by inserting explanations as footnotes to Table2.

---

## [Decision Letter · Decision Letter 1]

17 Mar 2020

The novel microRNAs hsa-miR-nov7 and hsa-miR-nov3 are over-expressed in locally advanced breast cancer

PONE-D-19-30537R1

Dear Dr. Knappskog,

We are pleased to inform you that your manuscript has been judged scientifically suitable for publication and will be formally accepted for publication once it complies with all outstanding technical requirements.

With kind regards,

Bernard Mari, Ph.D

Academic Editor

PLOS ONE

Additional Editor Comments (optional):

Reviewers' comments:

Reviewer's Responses to Questions

**Comments to the Author**

1. If the authors have adequately addressed your comments raised in a previous round of review and you feel that this manuscript is now acceptable for publication, you may indicate that here to bypass the “Comments to the Author” section, enter your conflict of interest statement in the “Confidential to Editor” section, and submit your "Accept" recommendation.

Reviewer #2: All comments have been addressed

2. Is the manuscript technically sound, and do the data support the conclusions?

Reviewer #2: Yes

3. Has the statistical analysis been performed appropriately and rigorously? 

Reviewer #2: Yes

4. Have the authors made all data underlying the findings in their manuscript fully available?

Reviewer #2: Yes

5. Is the manuscript presented in an intelligible fashion and written in standard English?

Reviewer #2: Yes

6. Review Comments to the Author

Reviewer #2: Authors have addressed all the queries raised and updated the manuscript accordingly. Revised manuscript may now be accepted for publication.

7. PLOS authors have the option to publish the peer review history of their article (what does this mean?). If published, this will include your full peer review and any attached files.

Reviewer #2: Yes: Manoj Kumar

---

## [Editor Report · Acceptance letter]

25 Mar 2020

PONE-D-19-30537R1 

The novel microRNAs hsa-miR-nov7 and hsa-miR-nov3 are over-expressed in locally advanced breast cancer 

Dear Dr. Knappskog:

I am pleased to inform you that your manuscript has been deemed suitable for publication in PLOS ONE. Congratulations! Your manuscript is now with our production department. 

With kind regards,

on behalf of

Dr. Bernard Mari 

Academic Editor

PLOS ONE